# RETHINKING SPIKING NEURAL NETWORKS FROM AN ENSEMBLE LEARNING PERSPECTIVE

**Yongqi Ding, Lin Zuo**[*]**, Mengmeng Jing, Pei He, Hanpu Deng**
School of Information and Software Engineering
University of Electronic Science and Technology of China

## ABSTRACT

Spiking neural networks (SNNs) exhibit superior energy efficiency but suffer from limited performance. In this paper, we consider SNNs as ensembles of temporal subnetworks that share architectures and weights, and highlight a crucial issue that affects their performance: excessive differences in initial states (neuronal membrane potentials) across timesteps lead to unstable subnetwork outputs, resulting in degraded performance. To mitigate this, we promote the consistency of the initial membrane potential distribution and output through membrane potential smoothing and temporally adjacent subnetwork guidance, respectively, to improve overall stability and performance. Moreover, membrane potential smoothing facilitates forward propagation of information and backward propagation of gradients, mitigating the notorious temporal gradient vanishing problem. Our method requires only minimal modification of the spiking neurons without adapting the network structure, making our method generalizable and showing consistent performance gains in 1D speech, 2D object, and 3D point cloud recognition tasks. In particular, on the challenging CIFAR10-DVS dataset, we achieved 83.20% accuracy with only four timesteps. This provides valuable insights into unleashing the potential of SNNs.

## 1 INTRODUCTION

As the third generation of neural networks, spiking neural networks (SNNs) transmit discrete spikes between neurons and operate over multiple timesteps (Maass, 1997). Benefiting from the low power consumption and spatio-temporal feature extraction capability, SNNs have achieved widespread applications in spatio-temporal tasks (Wang & Yu, 2024; Chakraborty et al., 2024). In particular, ultra-low latency and low-power inference can be achieved when SNNs are integrated with neuromorphic sensors and neuromorphic chips (Yao et al., 2023; Ding et al., 2024).

To advance the performance of SNNs, previous work has improved the training method (Wu et al., 2018; Bu et al., 2022; Zuo et al., 2024b), network architecture (Yao et al., 2023; Shi et al., 2024), and neuron dynamics (Taylor et al., 2023; Fang et al., 2021b; Ding et al., 2023) to significantly reduce the performance gap between SNNs and artificial neural networks (ANNs). Typically, these methods treat the spiking neurons in an SNN as an activation function that evolves over timesteps $T$, with the membrane potential expressing a continuous neuronal state. In this paper, we seek to rethink the spatio-temporal dynamics of SNNs from an alternative perspective: ensemble learning, and explore the key factor influencing the ensemble to optimize its performance.

For an SNN $f(\theta)$, its instance $f(\theta)_t$ produces an output $O_t$ at each timestep $t$, we consider this instance to be a temporal subnetwork. In this way, we can obtain a collection of $T$ temporal subnetworks with the same architecture $f(\cdot)$ and parameters $\theta$: $\{f(\theta)_1, f(\theta)_2, \cdots, f(\theta)_T\}$. In general, the final output of the SNN is the average of the outputs over $T$ timesteps: $O = \frac{1}{T}\sum_{t=1}^{T} O_t$. We view this averaging operation as an ensemble strategy: averaging the different outputs of multiple models improves overall performance (Rokach, 2010; Allen-Zhu & Li, 2020). From this perspective, we can consider an SNN as an ensemble of $T$ temporal subnetworks.

---

[*]Corresponding author: linzuo@uestc.edu.cn. Emails: Yongqi Ding (yqding@std.uestc.edu.cn), Mengmeng Jing (jingmeng1992@gmail.com), Pei He (hepei@std.uestc.edu.cn), Hanpu Deng (hp-deng@std.uestc.edu.cn).

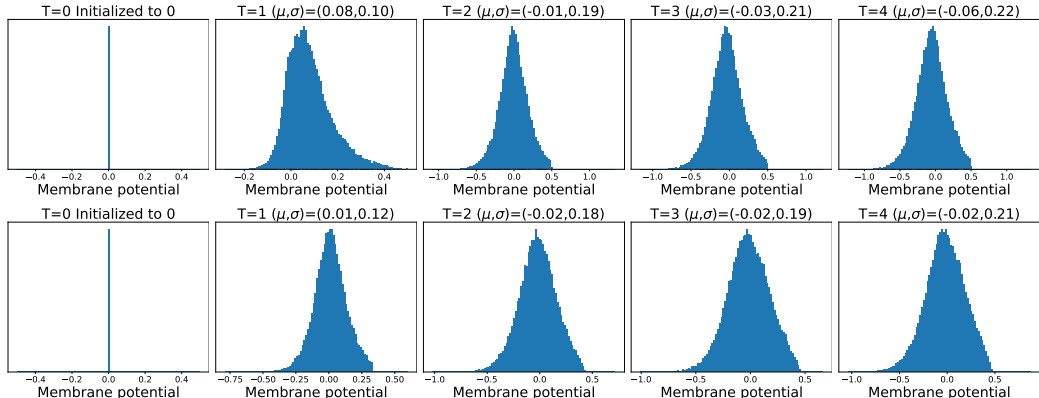

Figure 1: Membrane potential distribution on CIFAR10-DVS, where $\mu$ and $\sigma$ denotes the mean and standard deviation, respectively. **Top**: The membrane potential distribution of the vanilla SNN varies greatly across timesteps, which affects performance. **Bottom**: Our method allows for a more stable distribution with smaller differences across timesteps. See Appendix A.10 for more visualizations.

Temporal subnetworks share architecture and parameters, and their output variance arises from their different neuronal states, i.e., different membrane potentials $U(t)$ at each timestep trigger different output spikes $S(t)$. For example, for a leaky integrate-and-fire (LIF) neuron (Wu et al., 2018) (see Section 3.1) with an initial membrane potential of 0 and a time constant and firing threshold of 2.0 and 1.0, respectively, the neuron will produce spikes $\{0, 1, 1, 0, 1\}$ even if inputs of intensity 0.8 are repeated over 5 timesteps. In light of this, we have identified a factor that is usually overlooked, but has a major impact on the performance of this temporal subnetwork ensemble learning: *if the difference in membrane potentials across timesteps is too large, it will lead to unstable outputs and thus affect the ensemble performance.* In particular, the initial membrane potential is usually set to 0 (Wu et al., 2018; Ding et al., 2024), which leads to a drastic discrepancy in the membrane potential for the first two timesteps, thus degrading the performance of the SNN. To illustrate this phenomenon, we have visualized the membrane potential distribution in Fig 1(Top), where the distribution differences across timesteps can be clearly seen. In Table 1, we further explore the performance of the trained SNN for inference with 1 to 5 timesteps. The results show that the output of the vanilla SNN is poorly informative at the first timestep and only achieves decent performance after integrating subsequent temporal subnetworks with smaller differences in the membrane potential distribution. Additionally, the output is visualized in Fig. 4, again showing that the first two timestep outputs are confusing and thus affect the overall output. Therefore, to improve overall performance, the problem of excessive difference in membrane potential distribution and the resulting output should be mitigated rather than simply ignored.

To this end, we propose membrane potential smoothing to reduce the difference in membrane potential distribution and thus improve the overall ensemble performance. At each timestep, we adaptively smooth the membrane potentials using that of the previous timestep, pushing the initial states of these temporal subnetworks more consistent and preventing them

Table 1: Comparison of SNNs with 1 to 5 inference timesteps.

|  | T=1 | T=2 | T=3 | T=4 | T=5 |
|---|---|---|---|---|---|
| Vanilla SNN | 10.00 | 60.10 | 69.50 | 73.30 | 74.10 |
| Random MP | 11.90 | 61.50 | 71.40 | 72.30 | 74.90 |
| Ours | 66.60 | 74.30 | 75.50 | 75.70 | 76.60 |

from producing outputs with excessive variances. We visualize the smoothed membrane potential in Fig. 1(Bottom), where the obtained membrane potential is shown to be smoother than the vanilla SNN. Meanwhile, membrane potential smoothing creates new pathways for forward propagation of information and backward propagation of gradients (see Fig. 2), alleviating temporal gradient vanishing (Meng et al., 2023; Huang et al., 2024) and boosting the performance from another side. In addition, we propose to guide temporally adjacent subnetworks through distillation, encouraging them to produce more consistent outputs, further enhancing ensemble stability and performance. Notably, membrane potential smoothing employs only one additional smoothing coefficient, and the distillation guidance only modifies the training loss without affecting the architecture, making our method superiorly generalizable. Extensive experiments in 1D, 2D, and 3D scenarios demonstrate the effectiveness, generalizability, and performance advantages of our method. Our contribution can be summarized as follows:

- We consider the SNN as an ensemble of multiple temporal subnetworks and point out that excessive differences in membrane potential distributions across timesteps, and hence output instability, is the key factor affecting performance.

- We propose membrane potential smoothing and temporally adjacent subnetwork guidance to adaptively reduce membrane potential differences and enhance output consistency across timesteps, respectively, while facilitating the propagation of forward information and backward gradients to improve ensemble stability and overall performance.

- Extensive experiments on neuromorphic 1D speech/2D object recognition, and 3D point cloud classification tasks confirm the effectiveness, versatility, and performance advantages of our method. With only 4 timesteps, we achieved 83.20% accuracy on the challenging CIFAR10-DVS dataset.

## 2 RELATED WORK

### 2.1 SPIKING NEURAL NETWORK

Existing methods for training SNNs avoid the non-differentiability of the spiking neurons either by converting a pre-trained ANN (Rueckauer et al., 2017) or by using the surrogate gradient for direct training (Wu et al., 2018). Conversion-based methods require large latencies and struggle with the temporal properties of SNNs (Deng & Gu, 2021; Bu et al., 2022; kang you et al., 2024), the surrogate gradient-based methods are widely used as they can achieve decent performance with smaller latencies (Taylor et al., 2023; Zuo et al., 2024a; Hu et al., 2024). In addition to training methods, previous work has focused on improving network architectures and spiking neuron dynamics, such as the Spiking Transformer architecture (Yao et al., 2023; Shi et al., 2024), the ternary spike (Guo et al., 2024), and the attention spiking neuron (Ding et al., 2023). Compared to existing methods, we rethink the spatio-temporal dynamics of the SNN from the perspective of ensemble learning, identify the key factor affecting its performance: the excessive difference in membrane potential distribution, and propose solutions. Our solutions do not modify the core philosophies of these existing methods and are therefore compatible with a wide range of architectures and neuron types, and integration with existing methods can further unleash the potential of SNNs.

### 2.2 ENSEMBLE LEARNING

Ensemble learning aggregates the predicted outputs of multiple models to improve the performance of a deep learning model (Rokach, 2010; Allen-Zhu & Li, 2020). To reduce ensemble overhead, some methods use a backbone network and multiple heads to produce multiple outputs (Tran et al., 2020; Ruan et al., 2023), or use checkpoints during training for the ensemble (Furlanello et al., 2018; Lee et al., 2022). In the field of SNNs, previous studies have ensembled multiple SNN models to improve performance without optimizing the ensemble overhead (Neculae et al., 2021; Elbrecht et al., 2020; Panda et al., 2017; Hussaini et al., 2023). In this paper, we consider each timestep SNN instance as a temporal subnetwork and treat the entire SNN as an ensemble, thus avoiding additional ensemble overhead. A previous study (Ren et al., 2023) attributed the effectiveness of SNNs in static point cloud classification to the ensemble effect, without further analysis. Instead, we point out the key factor influencing the ensemble performance: excessive differences in membrane potential distributions can lead to unstable outputs of these subnetworks, and propose solutions to mitigate this problem, thereby improving the performance. Moreover, our experiments on various tasks suggest that this ensemble instability is ubiquitous and should be highlighted rather than simply ignored.

It is worth noting that previous ANN ensemble methods increase diversity within reasonable limits to promote generalization (Ali et al., 2021; Zhang et al., 2020). Instead, we take a different philosophy in SNNs, reducing difference rather than increasing diversity to promote stability, because the temporal subnetworks in SNNs are already beyond the limits of effective ensemble, and excessive diversity will only degrade overall performance. The necessity to reduce the cross-timestep differences of the SNN is discussed in detail in Appendix A.1.

### 2.3 TEMPORAL CONSISTENCY IN SNNS

Previous studies have shown that promoting temporal consistency can improve the performance of SNNs, such as distillation (Zuo et al., 2024a; Dong et al., 2024) and contrastive learning (Qiu et al., 2024) in the temporal dimension. However, existing methods directly promote output/feature

consistency, similar to ANNs, without adequately considering the properties of SNNs. In contrast to existing methods, this paper highlights the negative impact of differences in membrane potential distributions across timesteps from an ensemble perspective and proposes to improve distribution consistency. Compared to output/feature consistency, membrane potential distribution consistency offers significant performance gains and can be combined with them to synergistically maximize performance.

## 3 METHOD

In this section, we describe the temporal dynamics of the SNN from the basic LIF neuron model and interpret it as the ensemble of multiple temporal subnetworks. We then show that excessive differences in membrane potentials across timesteps affect ensemble performance, and we mitigate this problem by adaptively smoothing membrane potentials. In addition, we encourage temporally adjacent subnetworks to produce stable and consistent outputs through distillation, further facilitating ensemble stability and performance.

### 3.1 SNN AS AN ENSEMBLE OF TEMPORAL SUBNETWORKS

SNNs transmit information by generating binary spikes from spiking neurons. In this paper, we use the most commonly used LIF neuron model (Wu et al., 2018). LIF neurons continuously receive inputs from presynaptic neurons accumulating membrane potential $H$, and a spike $S$ is fired and resets the membrane potential when it reaches the firing threshold $\vartheta$. The LIF neuron dynamics can be expressed as:

$$H_i^l(t) = U_i^l(t) + I_i^l(t) = \left(1 - \frac{1}{\tau}\right) H_i^l(t-1) + I_i^l(t), \text{charge} \tag{1}$$

$$S_i^l(t) = \begin{cases} 1, & H_i^l(t) \geq \vartheta \\ 0, & H_i^l(t) < \vartheta \end{cases}, \text{fire spike} \tag{2}$$

$$H_i^l(t) = H_i^l(t) - S_i^l(t)\vartheta, \text{reset} \tag{3}$$

where $l$, $i$, and $t$ denote the layer, neuron, and timestep indexes, respectively, and $I_i^l(t) = \sum_j W_{i,j}^l S_j^{l-1}(t)$ is the cumulative current of the previous layer's neuron outputs and weights. $\tau$ is the time constant that controls the decay of the membrane potential with time. $U_i^l(t)$ is the initial membrane potential at timestep $t$.

From Eq. 1, we can see that the output $S_i^l(t)$ of the neuron at timestep $t$ depends on its initial membrane potential $U_i^l(t)$. The membrane potential evolves continuously, so the SNN output varies from timestep to timestep. Established methods recklessly compute the average over $T$ timesteps outputs $O = \frac{1}{T} \sum_{t=1}^{T} O_t$ without considering the rationale behind it, leading to suboptimal performance, whereas we explore this from an ensemble perspective to improve the performance of SNNs.

We consider each timestep of the SNN as a temporal subnetwork sharing the architecture $f(\cdot)$ and the parameter $\theta$, and different subnetworks produce different outputs $O_t$ arising from distinct neuron membrane potentials $U(t)$. When the membrane potential difference of these subnetworks is small, their outputs vary slightly, and the ensemble can promote the generalizability of the SNN. However, excessive differences in membrane potentials can lead to drastically different outputs from these subnetworks and degrade the SNN. Unfortunately, vanilla SNNs seem to suffer from this degradation, especially since the initial membrane potential is usually set to 0 (Wu et al., 2018; Ding et al., 2024), the membrane potentials of the first two timesteps show drastic differences, see Fig. 1. This membrane potential discrepancy leads to highly unstable outputs across timesteps, which increases the optimization difficulty and degrades the ensemble performance. In addition, we show the performance of the trained SNN with different inference timesteps in Table 1, and the results show that the output of the vanilla SNN at the first timestep is not discriminative at all.

### 3.2 MEMBRANE POTENTIAL SMOOTHING

To mitigate the above degradation problem, a natural solution is to randomly initialize the membrane potential to reduce the membrane potential difference for the first two timesteps, similar to (Ren et al., 2023). However, the experiments we show in Table 1 indicate that this practice (Random

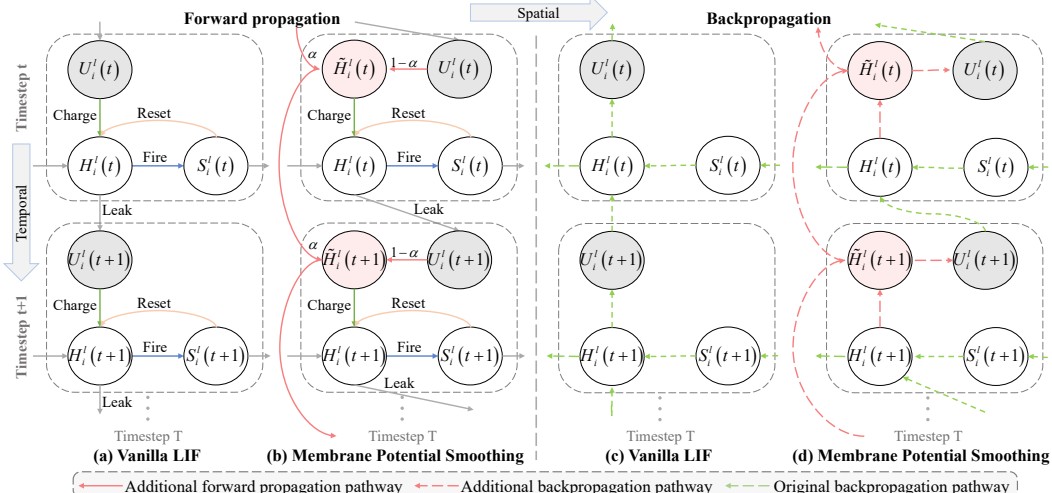

Figure 2: Illustration of (a) the vanilla LIF neuron and (b) the membrane potential smoothing. We smooth the membrane potential at timestep $t$ using the layer-shared coefficient $\alpha^l$ and the smoothed membrane potential $\tilde{H}_i^l(t-1)$ at timestep $t-1$ to reduce membrane potential differences and create additional information/gradient propagation pathways.

MP) only slightly alleviates the problem and still struggles to achieve satisfactory performance. To this end, we propose membrane potential smoothing, which mitigates degradation by adaptively reducing the membrane potential difference between adjacent timesteps using a learnable smoothing coefficient.

At each timestep, we consider the initial membrane potential state $U_i^l(t)$ of the spiking neuron as the initial state of the corresponding subnetwork. We argue that by allowing these subnetworks to have similar initial states, the output spikes generated after receiving the input current will also be similar, resulting in stable SNN outputs (The input currents typically follow the same distribution due to the normalization layer). Therefore, we weight the current initial state by the smoothed state of the previous timestep with a layer-shared smoothing coefficient $\alpha^l$ to reduce the difference between the two. The smoothed membrane potential $\tilde{H}_i^l(t)$ receives the input current from the previous layer, which then generates spikes and resets the membrane potential, iterating to the next timestep. The membrane potential smoothing and the charge dynamics of a spiking neuron can be expressed as:

$$\tilde{H}_i^l(t) = \alpha^l \tilde{H}_i^l(t-1) + (1-\alpha^l)U_i^l(t), \text{smoothing} \tag{4}$$

$$H_i^l(t) = \tilde{H}_i^l(t) + I_i^l(t).\text{charge} \tag{5}$$

The smoothing coefficient $\alpha^l$ and the parameter $\theta$ of the SNN are co-optimized during training to achieve the optimal smoothing effect. To ensure that $\alpha^l \in (0,1)$, in the practical implementation we train the parameter $\beta^l$ and let $\alpha^l = sigmoid(\beta^l)$. By default, $\beta^l$ is initialized to 0, i.e. the initial value of $\alpha^l$ is 0.5. In Section 4.3, we will analyze the influence of the initial value of $\alpha^l$ on the performance and convergence. Since the spike activity (Eq. 2) is not differentiable, we use the rectangular function (Wu et al., 2018) to calculate the spike derivative:

$$\frac{\partial S_i^l(t)}{\partial H_i^l(t)} \approx \frac{\partial h(H_i^l(t), \vartheta)}{\partial H_i^l(t)} = \frac{1}{a}\text{sign}(|H_i^l(t) - \vartheta| < \frac{a}{2}), \tag{6}$$

where $a$ is the hyperparameter that controls the shape of the rectangular function and is set to 1.0. Accordingly, the derivative of a spike with respect to $\alpha^l$ can be calculated as:

$$\frac{\partial S_i^l(t)}{\partial \alpha^l} = \frac{\partial S_i^l(t)}{\partial H_i^l(t)} \frac{\partial H_i^l(t)}{\partial \tilde{H}_i^l(t)} \frac{\partial \tilde{H}_i^l(t)}{\partial \alpha^l} \approx \frac{1}{a}\text{sign}(|H_i^l(t) - \vartheta| < \frac{a}{2})(\tilde{H}_i^l(t-1) - U_i^l(t)). \tag{7}$$

The derivative of the loss function $\mathcal{L}$ with respect to $\alpha^l$ can be calculated as:

$$\frac{\partial \mathcal{L}}{\partial \alpha^l} = \sum_t^T \sum_i^{n_l} \frac{\partial \mathcal{L}}{\partial S_i^l(t)} \frac{\partial S_i^l(t)}{\partial \alpha^l}, \tag{8}$$

where $n_l$ is the number of neurons in layer $l$.

It is worth noting that in addition to mitigating the difference in membrane potential distribution, membrane potential smoothing can also facilitate the propagation of the gradient in the temporal dimension. Previous studies have shown that in SNNs, the temporal gradient is a small percentage (Meng et al., 2023) and is highly susceptible to gradient vanishing, leading to performance degradation (Huang et al., 2024). As shown in Fig. 2, our method establishes a forward information transfer pathway from $\tilde{H}_i^l(t-1)$ to $\tilde{H}_i^l(t)$, and also propagates the error gradient in the backward direction, thus mitigating the influence of the temporal gradient vanishing (Detailed theoretical analysis is provided in Appendix A.3). From another perspective, the additional pathways can be viewed as residual connections in the temporal dimension, facilitating the propagation of information and gradients.

Notably, membrane potential smoothing is integrated with LIF neurons in this paper (see Appendix A.2 for complete neuron dynamics), but this smoothing method is not limited to specific neuron types and can be integrated with other pre-existing neurons to further improve performance (See Table 3).

### 3.3 TEMPORALLY ADJACENT SUBNETWORK GUIDANCE

Membrane potential smoothing aims to pull together the initial states of the subnetworks. In addition, we propose the temporally adjacent subnetwork guidance to further promote the output stability of these subnetworks and improve the ensemble performance.

Inspired by knowledge distillation (Hinton, 2015), we guide the output by identifying the "teacher" and "student" from two temporally adjacent subnetworks. Since spiking neurons need to accumulate membrane potentials before they can produce stable spiking outputs, we treat the early timestep subnetwork as a weak "student" that is guided by the stable "teacher" with a later timestep. Taking the $t$-th and $t+1$-th subnetworks as an example, the output probabilities are first calculated based on the logits $O_t$ and $O_{t+1}$ of the two subnetworks, respectively:

$$p(t) = \frac{e^{O_{t,j}/T_{KL}}}{\sum_{c=1}^{C} e^{O_{t,c}/T_{KL}}}, p(t+1) = \frac{e^{O_{t+1,j}/T_{KL}}}{\sum_{c=1}^{C} e^{O_{t+1,c}/T_{KL}}}, \tag{9}$$

where $C$ denotes the $C$-way classification task, the subscript $j$ indicates the $j$-th class, and $T_{KL}$ is the temperature hyperparameter set to 2. We then use KL divergence to encourage the output probability of subnetwork $t$ to be as similar as possible to the output probability of subnetwork $t+1$ (For the regression task of predicting continuous values, we compute the output MSE loss directly, as in the object detection task in Appendix A.7):

$$\mathcal{L}_t = T_{KL}{}^2 KL(p(t+1)||p(t)) = T_{KL}{}^2 \sum_{j=1}^{C} p(t+1)_j log(\frac{p(t+1)_j}{p(t)_j}). \tag{10}$$

By performing guidance between each pair of temporally adjacent subnetworks, we obtain a total of $T-1$ guidance losses: $\{\mathcal{L}_1, \mathcal{L}_2, \cdots, \mathcal{L}_{T-1}\}$. Instead of accumulating all these losses directly, we keep the largest one and drop the others with a probability $P$, as in (Sariyildiz et al., 2024). This promotes consistency while preserving diversity among subnetworks, rather than eliminating differences altogether, thus ensuring generalization of the ensemble. In this paper, $P$ is set to 0.5.

We define the function that selects the largest loss and randomly discards the others as $drop(\cdot)$ (see Appendix A.4 for pseudocode). During training, $drop(\{\mathcal{L}_1, \mathcal{L}_2, \cdots, \mathcal{L}_{T-1}\})$ and cross-entropy loss $\mathcal{L}_{CE}$ synergistically train the SNN:

$$\mathcal{L}_{total} = \gamma\mathcal{L}_{guidance} + \mathcal{L}_{CE} = \gamma drop(\{\mathcal{L}_1, \mathcal{L}_2, \cdots, \mathcal{L}_{T-1}\}) + \mathcal{L}_{CE}, \tag{11}$$

where $\gamma$ is the coefficient for controlling the guidance loss, which is set to 1.0 by default.

In this way, we synergistically increase the stability of the ensemble at both the level of the initial state (membrane potential) and the output of the subnetwork, greatly improving the overall performance of the SNN. We show the pseudocode for the training process of the temporally adjacent subnetwork guidance in Algorithm 1.

---

**Algorithm 1** Temporally adjacent subnetwork guidance for SNNs

---

**Input**: input data $x$, label $Y$.
**Parameter**: timestep $T$, Guidance loss coefficient $\gamma$.
**Output**: Trained SNN.

1: Initialize SNN $f(\cdot)$ with parameters $\theta$
2: // Optimization over multiple iterations
3: **for** $i = 1, 2, \cdots, i_{train}$ iterations **do**
4:     **for** $t = 1, 2, \cdots, T$ **do**
5:        $O_t = f(\theta; x(t))$ ; // Calculate the output at timestep $t$
6:        **if** $t > 1$ **then**
7:           $\mathcal{L}_{t-1} = \text{KL}(O_{t-1}; O_t)$ ; // Calculate the guidance loss by Eq. 10
8:        **end if**
9:     **end for**
10:    $O = \frac{1}{T}\sum_{t=1}^{T} O_t$ ; // Calculate the ensemble average output
11:    $\mathcal{L}_{CE} = \text{Cross-entropy}(O, Y)$ ; // Calculate the cross-entropy loss
12:    $\mathcal{L}_{guidance} = drop(\{\mathcal{L}_1, \mathcal{L}_2, \cdots, \mathcal{L}_{T-1}\})$ ; // Random drop guidance loss
13:    $\mathcal{L}_{total} = \gamma\mathcal{L}_{guidance} + \mathcal{L}_{CE}$ ; // Calculate total loss by Eq. 11
14:    Backpropagation and optimize model parameters $\theta$;
15: **end for**
16: **return** Trained SNN.

---

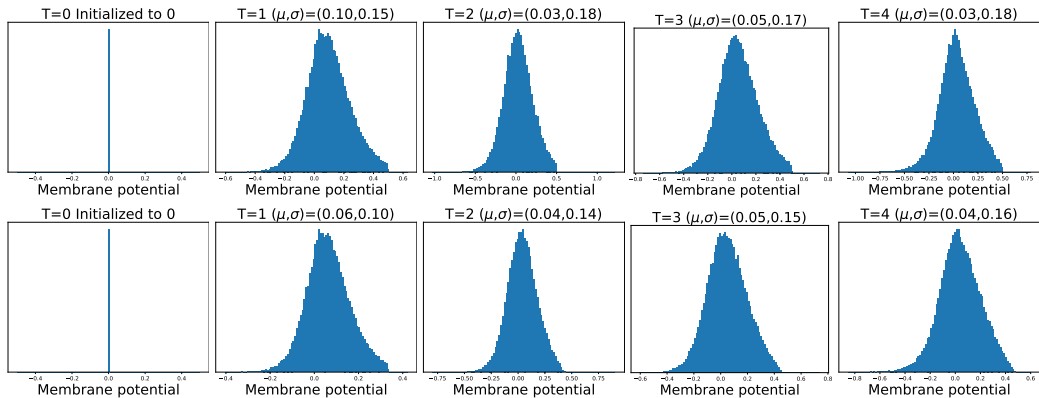

Figure 3: Visualization of the membrane potential distribution before (top) and after (bottom) smoothing. Smoothing reduces distribution differences, especially for $T1 \rightarrow T2 \rightarrow T3$.

## 4 EXPERIMENTS

We have conducted extensive experiments with various architectures (RNN, VGG, ResNet, Transformer, PointNet++) on neuromorphic speech (1D), object (2D) recognition, and 3D point cloud classification tasks. Please see the Appendix A.5 for detailed experimental setup.

### 4.1 ABLATION STUDY

We perform ablation studies using the VGG-9, ResNet-18, and SpikingResformer (Shi et al., 2024) architectures on the neuromorphic object recognition benchmark datasets CIFAR10-DVS (Li et al., 2017) and DVS-Gesture (Amir et al., 2017). The results in Table 2 show that excessive differences in

Table 2: Ablation study results of the proposed method (%).

| Dataset | Method | VGG-9 | ResNet-18 | SpikingResformer |
|---|---|---|---|---|
| CIFAR10-DVS | Baseline | 73.97 | 66.73 | 77.60 |
| | +Smooth | $74.80_{+0.83}$ | $68.07_{+1.34}$ | $78.45_{+0.85}$ |
| | +Guidance | $76.23_{+2.26}$ | $69.33_{+2.60}$ | $79.23_{+1.63}$ |
| | +Both | $\mathbf{76.77}_{+2.80}$ | $\mathbf{70.03}_{+3.30}$ | $\mathbf{80.60}_{+3.00}$ |
| DVS-Gesture | Baseline | 87.85 | 80.56 | 90.63 |
| | +Smooth | $89.93_{+2.08}$ | $82.99_{+2.43}$ | $91.32_{+0.69}$ |
| | +Guidance | $91.32_{+3.47}$ | $84.84_{+4.28}$ | $93.06_{+2.43}$ |
| | +Both | $\mathbf{93.23}_{+5.38}$ | $\mathbf{85.30}_{+4.74}$ | $\mathbf{94.44}_{+3.81}$ |

membrane potentials across timesteps are prevalent across different architectures and datasets, and that our solution consistently mitigates this problem. The visualization of the membrane potential distribution before and after smoothing is shown in Fig. 3, and it can be seen that smoothing does indeed reduce the distribution difference.

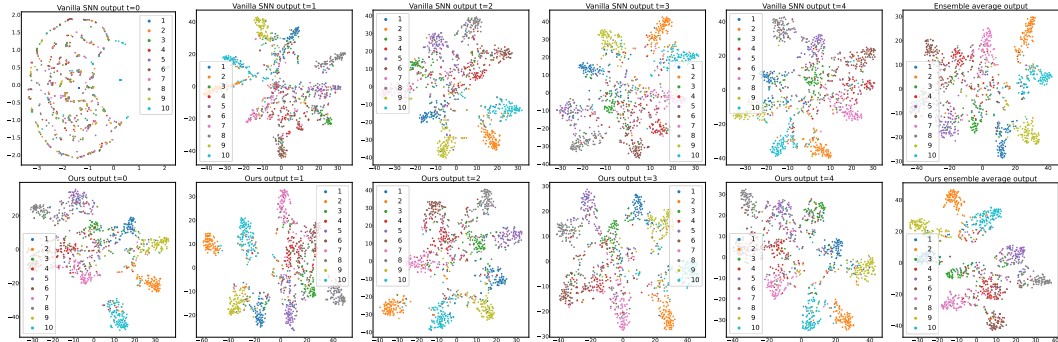

Figure 4: Two-dimensional t-SNE visualization on the CIFAR10-DVS dataset. **Top**: The output of the vanilla SNN varies greatly across timesteps, and the overall output is confusing, making it difficult to distinguish between classes. **Bottom**: The output of our SNN is more stable across timesteps and more distinguishable across classes, especially for the first two timesteps.

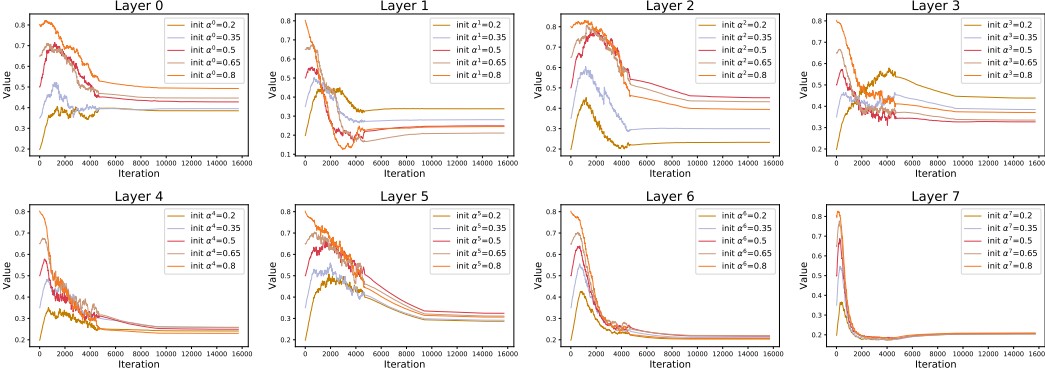

Figure 5: The optimization trend of $\alpha$ during training. $\alpha$ with different initial values gradually converge with training iterations, indicating that our method is insensitive to the initial value of $\alpha$.

In addition, the generalizability experiments with membrane potential smoothing for PLIF (Fang et al., 2021b) and CLIF (Huang et al., 2024) (on CIFAR10-DVS) as well as for RNN-LIF and DH-LIF (Zheng et al., 2024) (on the neuromorphic speech dataset SHD (Cramer et al., 2022)) are shown in Table 3. In particular,

Table 3: Generalizability experiments (%) of membrane potential smoothing on different neurons.

| | PLIF | CLIF | RNN-LIF | DH-LIF |
|---|---|---|---|---|
| Original | 74.83 | 74.97 | 81.87 | 89.86 |
| +Smooth | $75.10_{+0.27}$ | $75.97_{+1.00}$ | $83.04_{+1.17}$ | $90.33_{+0.47}$ |

we reduce rather than eliminate the differences between subnetworks, preserving the temporal properties of SNNs and thus providing performance gains even in time-dependent speech recognition tasks. Additional ablation experimental results and analysis are presented in Appendix A.6.

### 4.2 OUTPUT VISUALIZATION

We have visualized the output of the SNN in Fig. 4 with 2D t-distributed stochastic neighbor embedding (t-SNE) to show the improvement in output stability and distinguishability with our method. In Fig. 4(Top), the output of the vanilla SNN at each individual timestep varies widely, and the output of the first two timesteps in particular is confusing. This results in a poor distinguishability of its ensemble average output, which limits the recognition performance. Our SNN on the other hand, has more stable outputs across timesteps, and in particular the outputs generated at the first two timesteps are also well distinguished, as shown in Fig. 4(Bottom). Benefiting from the stability across timesteps, our final output is more spread across different classes of clusters and more compactly distributed within the clusters, yielding better performance. Please see Appendix A.10 for more visualization comparisons.

### 4.3 SMOOTHING COEFFICIENT ANALYSIS

By default, $\alpha$ is initialized to 0.5. To analyze the influence of the initialization value on $\alpha$, we visualize the optimization trend of $\alpha$ in each layer of VGG-9 trained on CIFAR10-DVS in Fig. 5,

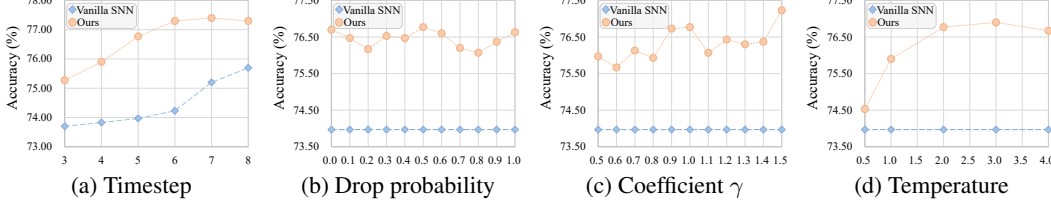

|                |                |                |                |
|:--------------:|:--------------:|:--------------:|:--------------:|
| (a) Timestep   | (b) Drop probability | (c) Coefficient $\gamma$ | (d) Temperature |

Figure 6: Influence of hyperparameters on performance. (a) Our method consistently outperforms vanilla SNN across different timesteps. (b) (c) Our method is insensitive to the drop probability of the guidance loss and the coefficient $\gamma$. (d) Our method is insensitive to the temperature hyperparameter within a reasonable range ($T_{KL} \geq 1$).

Table 4: Comparative results on neuromorphic datasets. *: self-implementation results with open-source code. †: knowledge transfer from static data. +: Data augmentation.

| Dataset | Method | Architecture | T | Accuracy (%) |
|---|---|---|---|---|
| CIFAR10-DVS | Ternary Spike (Guo et al., 2024) | ResNet-20 | 10 | 78.70 |
| | SLTT (Meng et al., 2023) | VGG-11 | 10 | 77.17 |
| | NDOT (Jiang et al., 2024a) | VGG-11 | 10 | 77.50 |
| | SSNN (Ding et al., 2024) | VGG-9 | 5 | 73.63 |
| | SLT (Anumasa et al., 2024) | VGG-9 | 5 | 74.23* |
| | CLIF (Huang et al., 2024) | VGG-9 | 5 | 74.97* |
| | SpikingResformer (Shi et al., 2024) | SpikingResformer-Ti | 5 | 77.60* |
| | SDT (Yao et al., 2023) | Spiking Transformer-2-256 | 5 | 72.53* |
| | TRR (Zuo et al., 2024b) | Spiking Transformer-2-256 | 5 | 75.55 |
| | **Ours** | VGG-9 | 5 | 76.77 |
| | | SpikingResformer-Ti | 5 | **80.60** |
| | | VGGSNN | 4 | **83.20**+ |
| DVS-Gesture | SSNN (Ding et al., 2024) | VGG-9 | 5 | 90.74 |
| | TRR (Zuo et al., 2024b) | VGG-9 | 5 | 91.67 |
| | SLT (Anumasa et al., 2024) | VGG-9 | 5 | 89.35* |
| | SpikingResformer (Shi et al., 2024) | SpikingResformer-Ti | 5 | 90.63* |
| | SDT (Yao et al., 2023) | Spiking Transformer-2-256 | 5 | 92.24* |
| | **Ours** | VGG-9 | 5 | 93.23 |
| | | SpikingResformer-Ti | 5 | **94.44** |
| N-Caltech101 | TCJA-TET-SNN (Zhu et al., 2024) | CombinedSNN | 14 | 82.50 |
| | EventMix (Shen et al., 2023) | ResNet-18 | 10 | 79.47 |
| | TIM (Shen et al., 2024) | Spikformer | 10 | 79.00 |
| | NDA (Li et al., 2022) | VGG-11 | 10 | 78.20 |
| | Knowledge-Transfer (He et al., 2024) | VGGSNN | 10 | 91.72*† |
| | SSNN (Ding et al., 2024) | VGG-9 | 5 | 77.97 |
| | TEBN (Duan et al., 2022) | VGG-9 | 5 | 81.24* |
| | **Ours** | VGG-9 | 5 | 82.71 |
| | | VGGSNN | 10 | **93.68**† |

where the initial values of $\alpha$ are taken from $\{0.2, 0.35, 0.5, 0.65, 0.8\}$. As can be seen in Fig. 5, $\alpha$ gradually converges at each layer during the training iterations. In particular, $\alpha$ in the last four layers gradually converges to almost the same optimal value, indicating that our method is robust to initial values. The first few layers of $\alpha$ do not converge to the same optimal value, which we attribute to the accumulation of errors in the surrogate gradient. The SNN uses the surrogate gradient instead of the derivative of the spike activity, which introduces a gradient error in backpropagation, and this error accumulates the further ahead the layer is. Eventually, these gradient errors cause the parameters of the earlier layers to be underoptimized, so that $\alpha$ does not converge to the same optimal value. However, compared to the 73.97% accuracy of the vanilla SNN, these different $\alpha$ initialization values achieved average accuracies of 76.20%, 76.80%, 76.77%, 75.75%, and 76.85%, respectively, both of which provide significant performance gains.

## 4.4 INFLUENCE OF HYPERPARAMETERS

To explore the influence of hyperparameters on the performance of the proposed method, we show in Fig. 6 the performance of VGG-9 on CIFAR10-DVS with different hyperparameter settings. We increased the timestep from 3 to 8 and found that the overall performance of the model also gradually increased and then saturated, and the average accuracy reached a maximum of 77.4% when the timestep was 7, as shown in Fig. 6(a). Compared to the vanilla SNN, our method consistently shows better performance.

The influence of the drop probability of the guidance loss and the coefficient $\gamma$ on the performance is shown in Fig. 6(b) and Fig. 6(c), and the results show that our method is not sensitive to these hyperparameters and consistently outperforms the vanilla SNN. This indicates that our method can significantly improve model performance without intentionally adjusting the hyperparameters.

In Fig. 6(d), we investigate the influence of the temperature hyperparameter $T_{KL}$ in the guidance on the performance, taking values set to $\{0.5, 1, 2, 3, 4\}$. The results show that the performance of our method fluctuates only slightly when $T_{KL} > 1$, and degrades when $T_{KL} = 0.5$ (still outperforming the vanilla SNN). We argue that this is due to the fact that too small a $T_{KL}$ causes the softened subnetwork logit to be too sharp, making it difficult to pull together subnetwork outputs that are already too dissimilar. These experiments show that our method is not sensitive to specific values as long as the hyperparameters are within reasonable ranges, and thus offers great robustness.

## 4.5 COMPARISON WITH EXISTING METHODS

**1D neuromorphic speech recognition.** Time-dependent speech recognition tasks require models with complex temporal processing capabilities, so stability across timesteps is extremely important. The comparative results with other SNNs on the SHD dataset are shown in Table 5. Our

Table 5: Comparative results on the 1D SHD dataset.

| Method | Architecture | Param (M) | Acc. (%) |
|---|---|---|---|
| Origin (Cramer et al., 2022) | LSTM | 0.43 | 89.20 |
| TIM (Shen et al., 2024) | Spikformer | - | 86.30 |
| TC-LIF (Zhang et al., 2024a) | Recurrent | 0.39 | 88.91[*] |
| DH-LIF (Zheng et al., 2024) | Recurrent | 0.39 | 89.86 |
| **Ours** | Recurrent | 0.39 | **91.19** |

method achieves a recognition accuracy of 91.19% with a lightweight RNN architecture, outperforming other methods. This confirms the effectiveness of our method on long time sequences.

**2D neuromorphic object recognition.** As shown in Table 4, our VGG-9 achieves an accuracy of 76.77% and 93.23% on CIFAR10-DVS and DVS-Gesture, respectively, with only 5 timesteps. Using the Transformer architecture, we achieved accuracies of 80.60% and 94.44%, respectively, significantly exceeding other methods. Specifically, with data augmentation, we were able to achieve 83.20% accuracy on the CIFAR10-DVS with 4 timesteps. In addition, we conducted experiments on N-Caltech101 (Orchard et al., 2015) and achieved 82.71% accuracy using VGG-9. Note that we only used vanilla LIF neurons and the general training method, which can further improve the performance when combined with other methods. For example, employing the knowledge transfer strategy (He et al., 2024), we were able to increase our accuracy to 93.68%.

**3D point cloud classification.** For the challenging point cloud classification, we performed experiments on the ModelNet10/40 datasets (Wu et al., 2015) using the lightweight PointNet++ architecture (Qi et al., 2017), and the comparative results are shown in Table 6. With $T = 2$,

Table 6: Comparative results (%) on point cloud classification.

| Method | Type | T | ModelNet10 | ModelNet40 |
|---|---|---|---|---|
| PointNet++ (Qi et al., 2017) | ANN | - | 95.50 | 92.16 |
| Converted SNN (Lan et al., 2023) | SNN | 16 | 92.75 | 89.45 |
| Spiking PointNet (Ren et al., 2023) | SNN | 2 | 92.98 | 88.46 |
| P2SResLNet (Wu et al., 2024) | SNN | 1 | - | 89.20 |
| **Ours** | SNN | 2 | **94.54** | **91.13** |
| | | 1 | 94.39 | 89.82 |

our method achieves 94.54% and 91.13% accuracy, respectively, outperforming other SNN models. To evaluate the one-timestep inference performance, we directly infer the trained two-timestep model with one timestep. This preserves the benefits of our method while avoiding additional training overhead. The results show that even with only one timestep, we can achieve accuracies of 94.39% and 89.82%, which still outperform other SNNs.

## 5 CONCLUSION

In this paper, we highlight a key factor that is generally overlooked in SNNs: excessive differences in membrane potential distributions can lead to unstable outputs across timesteps, thereby affecting performance. To mitigate this, we propose membrane potential smoothing and temporally adjacent subnetwork guidance to facilitate consistency of initial membrane potentials and outputs, respectively, thereby improving stability and performance. Meanwhile, membrane potential smoothing also facilitates the propagation of forward information and backward gradients, mitigating the temporal gradient vanishing and providing dual gains. Extensive experiments on neuromorphic speech (1D)/object (2D) recognition and 3D point cloud classification tasks confirmed the effectiveness and versatility of our method. We expect that our work will inspire the community to further analyze the spatio-temporal properties of SNNs.

ACKNOWLEDGMENTS

This work was supported by the National Natural Science Foundation of China under Grant No. 62276054 and 62406060.

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

# A APPENDIX

The Appendix section provides detailed theoretical analysis, neural dynamics, and additional experiments described below:

- Appendix A.1 details and experimentally confirms the necessity to reduce the variance across timesteps in SNNs.
- Appendix A.2 shows the dynamics of LIF neurons with integrated membrane potential smoothing.
- Appendix A.3 analyzes the effectiveness of membrane potential smoothing to mitigate the temporal gradient vanishing.
- Appendix A.4 provides pseudo-code for randomly dropping guidance losses and temporally adjacent subnetwork guidance.
- Appendix A.5 provides experimental details such as the dataset and training setup.
- Appendix A.6 shows the additional ablation studies and analysis of our method.
- Appendix A.7 demonstrates the application of the temporally adjacent subnetwork guidance to the object detection task.
- Appendix A.8 presents the analysis of the effectiveness and power consumption of our method for the spiking Transformer model with different timesteps.
- Appendix A.9 provides additional experiments on the static image datasets.
- Appendix A.10 shows additional visualizations to demonstrate the effectiveness of our method in promoting consistency in the membrane potential distribution.

A.1 THE NECESSITY TO REDUCE CROSS-TIMESTEP DIFFERENCES IN SNNs

Previous research on ensemble learning has shown that increasing the diversity of members in an ensemble contributes to overall generalization under certain constraints (Krogh & Vedelsby, 1994; Frankle et al., 2020; Zhang et al., 2020). In contrast, we point out that the differences in the output of the SNN across timesteps should be reduced, which at first glance seems to contradict the idea of ensemble learning. In this section, we will elucidate the necessity to reduce the differences in SNNs across timesteps by analyzing the constraints of ensemble learning and experimental results.

Previous studies have shown that the solutions learned by Stochastic Gradient Descent (SGD) lie on a nonlinear manifold (Draxler et al., 2018; Garipov et al., 2018; Nguyen, 2019) and that converged models with a common initial optimization path are linearly connected with a low-loss barrier (Frankle et al., 2020). Multiple optimal solutions that are linearly connected belong to a common *basin* that is separated from other regions of the loss landscape (Jain et al., 2023). The linearly connected multiple models in a common basin can thus be ensembled for additional gains. From this we derive a constraint that is important for ensemble, but typically overlooked: **The members of the ensemble should be converged optimal solutions, or at the very least optimized feasible solutions.** However, it is possible that a particular temporal submodel in the SNN is not a feasible solution. For example, the results in Table 1 show that the SNN submodel is only 10% accurate at the first timestep on the ten-class CIFAR10-DVS dataset, which is equivalent to a randomly initialized network that is far from an optimal or feasible solution. At this point, it has stepped outside the constraints of the ensemble member, so it cannot continue to increase the diversity of its members as it would in normal practice. Instead, we can make the infeasible solution gradually converge to the feasible solution by decreasing the difference between this overly outlier submodel (or infeasible solution) and the other submodels (feasible solutions). Finally, these feasible solutions can be used as an effective ensemble to improve overall stability and performance.

From another point of view, it is difficult to have both performance and diversity of tasks in an ensemble. To balance performance and diversity, existing methods use multiple losses during training to guide the optimization of the neural network ensemble (Ali et al., 2021; Zhang et al., 2020). Following (Zhang et al., 2020), we divide the training loss of an ensemble into three parts (Take the classification task as an example): i) the cross-entropy loss of ensemble members and labels; ii) the diversity of ensemble members; and iii) the aggregated loss of the ensemble.

- The cross-entropy loss of ensemble members and labels can be calculated as

$$\mathcal{L}_s = \mathcal{H}(q(\mathcal{T}_i(\mathbf{X})), \mathbf{Y}), \tag{12}$$

  where $\mathbf{X}$ is the input, $\mathbf{Y}$ is the label, $\mathcal{T}_i$ is the $i$-th member model, $q(\cdot)$ is the normalization function, and $\mathcal{H}$ is the cross entropy.

- The diversity of ensemble members can be calculated as

$$\mathcal{L}_d = 1 - \frac{1}{N} \sum_{1 \le i \ne j \le N} q\left(\mathcal{T}_i(\mathbf{X})\right) q\left(\mathcal{T}_j(\mathbf{X})\right), \tag{13}$$

  where $N$ is the number of members in the ensemble.

- The aggregated loss of the ensemble can be calculated as

$$\mathcal{L}_a = \mathcal{H}(\sum_i^N \gamma_i \cdot \mathcal{T}_i(\mathbf{X}), \mathbf{Y}), \tag{14}$$

  where $\gamma_i$ is the integration weight, bounded by $\sum_i^N \gamma_i = 1$.

The total loss during training is

$$\mathcal{L}_{ensemble} = \mathcal{L}_s + \mathcal{L}_a - \alpha \mathcal{L}_d, \tag{15}$$

where $\alpha$ is the coefficient controlling the increase in diversity that can be adaptively updated in (Zhang et al., 2020) with training iterations.

From Eq. 15, we can see that **ensemble learning cannot blindly increase the diversity of its members, but must strike a balance between overall performance and diversity**. Our method,

Table 7: Comparative results ($\times 10^{-3}$) of ensemble metrics on the CIFAR10-DVS dataset.

| | $\mathcal{L}_s \downarrow$ | $\mathcal{L}_d \uparrow$ | $\mathcal{L}_a \downarrow$ | $\mathcal{L}_{ensemble} \downarrow$ |
|---|---|---|---|---|
| Vanilla SNN | 25.1526 | **60.0711** | 13.6560 | -21.2624 |
| Ours | **16.2025** ($\downarrow 35.58\%$) | 58.2364 ($\downarrow 3.05\%$) | **12.2561** ($\downarrow 10.25\%$) | **-29.7778** ($\downarrow 40.05\%$) |

on the other hand, takes this into account by pointing out that excessive variance in the SNN (over-diversity) can degrade overall performance, thus requiring an appropriate reduction in diversity to achieve a better balance between performance and diversity.

To further confirm the effectiveness of our method in promoting a balance between overall performance and diversity, a comparison of the ensemble metrics on the CIFAR10-DVS dataset is presented in Table 7. The results show that although our method slightly reduces the diversity of the individual timestep submodels of the SNN, it achieves better performance in all other three metrics. In particular, our diversity is degraded by only a negligible 3.05% compared to the vanilla SNN, but the final integration loss is reduced by 40.05%, significantly improving the overall performance.

## A.2 INTEGRATING MEMBRANE POTENTIAL SMOOTHING IN LIF NEURONS

In this paper, membrane potential smoothing is integrated into commonly used LIF neurons. The neuron dynamics after integration can be expressed as:

$$U_i^l(t) = \left(1 - \frac{1}{\tau}\right) H_i^l(t-1), \text{leakage} \tag{16}$$

$$\tilde{H}_i^l(t) = \alpha^l \tilde{H}_i^l(t-1) + (1 - \alpha^l) U_i^l(t), \text{smoothing} \tag{17}$$

$$H_i^l(t) = \tilde{H}_i^l(t) + I_i^l(t), \text{charge} \tag{18}$$

$$S_i^l(t) = \begin{cases} 1, & H_i^l(t) \geq \vartheta \\ 0, & H_i^l(t) < \vartheta \end{cases}, \text{fire spike} \tag{19}$$

$$H_i^l(t) = H_i^l(t) - S_i^l(t)\vartheta, \text{reset} \tag{20}$$

where $\tilde{H}$ is the smoothed membrane potential. Note that $\tilde{H}(t-1)$ does not exist when $t = 0$, at which point the dynamics of the neuron are the same as the original LIF dynamics. When $t = 1$, $\tilde{H}(t-1) = \tilde{H}(0)$ is set to the initialized value of membrane potential $H$, which is 0 in this paper.

A.3  MEMBRANE POTENTIAL SMOOTHING MITIGATES TEMPORAL GRADIENT VANISHING

When training SNNs directly with the surrogate gradient, the gradient propagates backward in both spatial and temporal domains. Taking the vanilla LIF neuron as an example, we convert Eq. 1 to Eq. 3 into the following form to illustrate the temporal gradient vanishing problem:

$$\mathbf{H}^l(t) = (1 - \frac{1}{\tau})(\mathbf{H}^l(t-1) - \vartheta \mathbf{S}^l(t-1)) + \mathbf{W}^l \mathbf{S}^{l-1}(t), \tag{21}$$

The gradient of the loss $\mathcal{L}$ with respect to the weights $\mathbf{W}^l$ over $T$ timesteps is:

$$\nabla_{\mathbf{W}^l} \mathcal{L} = \sum_{t=0}^{T-1} \frac{\partial \mathcal{L}}{\partial \mathbf{H}^l(t)}^\top \mathbf{S}^{l-1}[t]^\top, \; l = L, L-1, \cdots, 1, \tag{22}$$

where $L$ is the total number of layers in the SNN.

The derivative of the loss $\mathcal{L}$ with respect to the membrane potential $\mathbf{H}^l(t)$ is:

$$\frac{\partial \mathcal{L}}{\partial \mathbf{H}^l(t)} = \frac{\partial \mathcal{L}}{\partial \mathbf{S}^l(t)} \frac{\partial \mathbf{S}^l(t)}{\partial \mathbf{H}^l(t)} + \sum_{t'=t+1}^{T-1} \frac{\partial \mathcal{L}}{\partial \mathbf{S}^l(t')} \frac{\partial \mathbf{S}^l(t')}{\partial \mathbf{H}^l(t')} \prod_{t''=1}^{t'-t} \epsilon^L(t'-t''), \; l = L, \tag{23}$$

$$\frac{\partial \mathcal{L}}{\partial \mathbf{H}^l(t)} = \frac{\partial \mathcal{L}}{\partial \mathbf{H}^{l+1}(t)} \frac{\partial \mathbf{H}^{l+1}(t)}{\partial \mathbf{S}^l(t)} \frac{\partial \mathbf{S}^l(t)}{\partial \mathbf{H}^l(t)} + \sum_{t'=t+1}^{T-1} \frac{\partial \mathcal{L}}{\partial \mathbf{H}^{l+1}(t')} \frac{\partial \mathbf{H}^{l+1}(t')}{\partial \mathbf{S}^l(t')} \frac{\partial \mathbf{S}^l(t')}{\partial \mathbf{H}^l(t')} \prod_{t''=1}^{t'-t} \epsilon^L(t'-t''), \; l < L, \tag{24}$$

where the blue portion on the left indicates the spatial gradient and the green portion on the right indicates the temporal gradient. $\epsilon^L(t)$ is defined as the sensitivity of the membrane potential $\mathbf{H}^l(t+1)$ to $\mathbf{H}^l(t)$ in adjacent timesteps (Meng et al., 2023; Huang et al., 2024):

$$\epsilon^l(t) \triangleq \frac{\partial \mathbf{H}^l(t+1)}{\partial \mathbf{H}^l(t)} + \frac{\partial \mathbf{H}^l(t+1)}{\partial \mathbf{S}^l(t)} \frac{\partial \mathbf{S}^l(t)}{\partial \mathbf{H}^l(t)}. \tag{25}$$

As can be seen in Eq. 25, the sensitivity $\epsilon^L(t)$ controls the percentage of the temporal gradient in the total gradient. However, for typical surrogate gradient settings, the diagonal matrix $\prod_{t''=1}^{t'-t} \epsilon^l(t'-t'')$ has only a small spectral norm (Meng et al., 2023). To illustrate, for the rectangular function used in Eq. 6, when $a = \vartheta$, the diagonal elements of $\epsilon^l(t)$ are

$$\epsilon^l(t)_{jj} = \begin{cases} 0, & \frac{1}{2}\vartheta < H_j^l(t) < \frac{3}{2}\vartheta, \\ 1 - \frac{1}{\tau}, & \text{otherwise.} \end{cases} \tag{26}$$

Typically, $1 - \frac{1}{\tau}$ is set to values less than 1, such as 0.5 and 0.25 (Ding et al., 2024; Guo et al., 2022), which causes the diagonal values of the matrix $\prod_{t''=1}^{t'-t} \epsilon^l(t'-t'')$ to become smaller as $t'-t$ increases, eventually causing the **temporal gradient vanishing** and degrading performance.

When membrane potential smoothing is used, the neuron at each timestep is smoothed by the smoothed membrane potential of the previous timestep, and the modified LIF neuron dynamics are shown in Appendix A.2. When $t > 0$, the sensitivity of the membrane potential at timestep $t+1$ with respect to the membrane potential at timestep $t$ can be calculated as:

$$\tilde{\epsilon}^l(\Delta t = 1) = \frac{\partial \mathbf{H}^l(t+1)}{\partial \tilde{\mathbf{H}}^l(t+1)} \frac{\partial \tilde{\mathbf{H}}^l(t+1)}{\partial \mathbf{H}^l(t)} = (1-\alpha)\epsilon^l(t). \tag{27}$$

When the timestep interval is 2, the sensitivity of the temporal gradient is:

$$\begin{aligned} \tilde{\epsilon}^l(\Delta t = 2) &= \frac{\partial \mathbf{H}^l(t+2)}{\partial \tilde{\mathbf{H}}^l(t+2)} \Big( \frac{\partial \tilde{\mathbf{H}}^l(t+2)}{\partial \tilde{\mathbf{H}}^l(t+1)} \frac{\partial \tilde{\mathbf{H}}^l(t+1)}{\partial \mathbf{H}^l(t)} + \frac{\partial \tilde{\mathbf{H}}^l(t+2)}{\partial \mathbf{H}^l(t+1)} \tilde{\epsilon}^l(\Delta t = 1) \Big) \\ &= \alpha(1-\alpha)\epsilon^l(t) + (1-\alpha)\epsilon^l(t)\tilde{\epsilon}^l(\Delta t = 1) \\ &= (\alpha + (1-\alpha)\epsilon^l(t))\tilde{\epsilon}^l(\Delta t = 1). \end{aligned} \tag{28}$$

By iterating over timestep according to the chain rule, the sensitivity of the temporal gradient can be obtained as

$$\tilde{\epsilon}^l(\Delta t) = (1-\alpha)\epsilon^l(t)(\alpha + (1-\alpha)\epsilon^l(t))^{\Delta t - 1} \tag{29}$$

Thus, the derivative of the loss $\mathcal{L}$ with respect to the membrane potential $\mathbf{H}^l(t)$ from Eq. 23 and Eq. 24 becomes

$$\frac{\partial \mathcal{L}}{\partial \mathbf{H}^l(t)} = \frac{\partial \mathcal{L}}{\partial \mathbf{S}^l(t)} \frac{\partial \mathbf{S}^l(t)}{\partial \mathbf{H}^l(t)} + \sum_{t'=t+1}^{T-1} \frac{\partial \mathcal{L}}{\partial \mathbf{S}^l(t')} \frac{\partial \mathbf{S}^l(t')}{\partial \mathbf{H}^l(t')} \tilde{\epsilon}^l(\Delta t = t' - t), \, l = L, \quad (30)$$

$$\frac{\partial \mathcal{L}}{\partial \mathbf{H}^l(t)} = \frac{\partial \mathcal{L}}{\partial \mathbf{H}^{l+1}(t)} \frac{\partial \mathbf{H}^{l+1}(t)}{\partial \mathbf{S}^l(t)} \frac{\partial \mathbf{S}^l(t)}{\partial \mathbf{H}^l(t)} + \sum_{t'=t+1}^{T-1} \frac{\partial \mathcal{L}}{\partial \mathbf{H}^{l+1}(t')} \frac{\partial \mathbf{H}^{l+1}(t')}{\partial \mathbf{S}^l(t')} \frac{\partial \mathbf{S}^l(t')}{\partial \mathbf{H}^l(t')} \tilde{\epsilon}^l(\Delta t = t' - t), \, l < L, \quad (31)$$

It can be seen that the sensitivity of the temporal gradient in the neuron after smoothing the membrane potential becomes $\tilde{\epsilon}^l(\Delta t = t' - t)$ instead of $\prod_{t''=1}^{t'-t} \epsilon^l(t' - t'')$. If the time step interval is small, such as 1, then $\tilde{\epsilon}^l(\Delta t = t' - t)$ is slightly lower than $\prod_{t''=1}^{t'-t} \epsilon^l(t' - t'')$, but this does not matter because the temporal gradient does not vanish at this point. If the time step interval is large, then the diagonal values in $\prod_{t''=1}^{t'-t} \epsilon^l(t' - t'')$ are greatly reduced, causing performance degradation. In contrast, $\tilde{\epsilon}^l(\Delta t = t' - t)$ can produce larger values at this point to mitigate the temporal gradient vanishing. For example, if $1 - \frac{1}{\tau} = 0.5$, $\alpha = 0.25$, and the timestep interval $\Delta t$ is 3, the valid value of the diagonal element in $\prod_{t''=1}^{t'-t} \epsilon^l(t' - t'')$ is 0.125, while the valid value in $\tilde{\epsilon}^l(\Delta t = t' - t)$ is 0.146. When the timestep interval is increased to 5, the difference between the two becomes 0.03125 versus 0.05722, at which point the sensitivity of the temporal gradient increases by 83%. This demonstrates the effectiveness of membrane potential smoothing in mitigating the temporal gradient vanishing.

### A.4 PYTORCH-STYLE CODE

For reproducibility, we provide Pytorch-style code for the drop function $drop(\cdot)$, which randomly drops guidance losses in the Algorithm 2 inspired by (Sariyildiz et al., 2024).

---

**Algorithm 2** PyTorch-style code for randomly dropping guidance losses

---

```
# losses: Guidance loss between T-1 temporally adjacent sub-networks.
# losses.shape: [T-1]
# p: random discard probability.
def drop(losses,p):
    T,B,C,H,W = x.shape
    w = torch.ones((losses.shape[0]))
    index = torch.argmax(losses)
    for i in range(losses.shape[0]):
        if i == index:
            continue
        else:
            p = random.random()
            if p < prob:
                w[i] = 0
    w.div_(w.sum())
    return w * kd_loss
```

---

## A.5 EXPERIMENTAL DETAILS

### A.5.1 DATASETS

In this paper, we perform experiments on the neuromorphic datasets CIFAR10-DVS, DVS-Gesture, and N-Caltech101, as well as the static image datasets CIFAR10, CIFAR100, and the 3D point cloud datasets ModelNet10 and ModelNet40.

CIFAR10-DVS (Li et al., 2017) is a benchmark dataset for neuromorphic object recognition, which contains 10,000 event samples of size $128 \times 128$. The dimension of each event sample $x$ is $[t, p, x, y]$, where $t$ is the time stamp, $p$ is the polarity of the event, indicating the increase or decrease of the pixel value, and $[x, y]$ are the spatial coordinates. There are 10 classes of samples in CIFAR10-DVS, and we divide each class of samples into training and test sets in the ratio of 9:1 to evaluate the model performance, the same as the existing work (Zuo et al., 2024b; Shi et al., 2024; Yao et al., 2023).

DVS-Gesture (Amir et al., 2017) dataset contains event samples for 11 gestures, of which 1176 are used for training and 288 are used for testing. The dimension of each event sample is $[t, p, x, y]$ and the spatial dimension size is $128 \times 128$.

N-Caltech101 (Orchard et al., 2015) contains event stream data for 101 objects, each sample with a spatial size of $180 \times 240$. There are 8709 samples in N-Caltech101, and we divide the training set and the test set at a ratio of 9:1.

Since the event data is of high temporal resolution, we use the SpikingJelly (Fang et al., 2023a) framework to integrate each event sample into $T$ event frames, where $T$ corresponds to the timestep of the SNN. In addition, event frames are downsampled to a spatial size of $48 \times 48$ before being input to the SNN.

The Spiking Heidelberg Digits (SHD) (Cramer et al., 2022) dataset contains 1000 spoken digits in 20 categories (from 0 to 9 in English and German) for the speech recognition task. For processing the SHD dataset, we followed (Zheng et al., 2024).

ModelNet10 (Wu et al., 2015) and ModelNet40 (Wu et al., 2015) are benchmark datasets for 3D point cloud classification. ModelNet10 contains point cloud data for 4899 objects in ten categories; ModelNet40 contains data for 12311 objects in 40 categories. For point cloud data preprocessing, we follow (Ren et al., 2023). We uniformly sample 1024 points on the mesh faces and input them into the SNN after normalizing them to a unit sphere.

### A.5.2 TRAINING SETTING

Our experiments are based on the PyTorch package, using the Nvidia RTX 4090 GPU. By default, we use the VGG-9 (Zuo et al., 2024b) architecture on the neuromorphic datasets. For ResNet-18, we use the architecture settings described in (Ding et al., 2024). We trained the model for 100 epochs using a stochastic gradient descent optimizer with an initial learning rate of 0.1 and a tenfold decrease every 30 epochs. We trained the VGG-9 and ResNet-18 models without using any data augmentation techniques, and the weight decay value was 1e-3. The batch size during training is 64. The firing threshold $\vartheta$ and membrane potential time constant $\tau$ of spiking neurons were 1.0 and 2.0, respectively.

When our method is used for the Transformer architecture SNN, we use the SpikingResformer (Shi et al., 2024) architecture. At this point, our training strategy is exactly the same as in the original paper, and we use the officially released training code directly.

Our method can be combined with the knowledge transfer strategy (He et al., 2024) to maximize performance gains, and we conducted experiments on N-Caltech101 using the officially released code. At this point, our training strategy is exactly the same as (He et al., 2024), but we set the batch size to 48 to reduce the memory overhead.

For the 1D SHD speech recognition task, we follow the training strategy and architecture of (Zheng et al., 2024). We use the one-layer two-branch recurrent network architecture defined in (Zheng et al., 2024) to which the proposed method is applied.

For the 3D point cloud classification task, we use the spiking version of the lightweight Point-Net++ (Qi et al., 2017) architecture. We directly use the code released by (Ren et al., 2023), and thus our training strategy is exactly the same as that of (Ren et al., 2023).

To reduce randomness, we report the average results of three independent experiments in our experiments, in addition to the experiments performed on Table 1 (randomly selected individual models) and on the large-scale ImageNet.

## A.6   ADDITIONAL ABLATION STUDY

In this section, we evaluate the proposed method on the neuromorphic speech recognition dataset SHD (Cramer et al., 2022). Taking DH-LIF (Zheng et al., 2024) as the baseline, the experimental results of the proposed method for ablation are shown in Table 8. The results show that our proposed method still improves the performance of the baseline in speech recognition tasks, demonstrating the generality of our method.

Furthermore, the experiments show that this view of an SNN as an ensemble of multiple subnetworks also holds for time-dependent tasks. Although time-dependent tasks require more time-varying information than static tasks, too much variance can still negatively affect their performance. Our method reduces excessive variance across timesteps, improving ensemble stability and overall performance. It is worth noting that our method does not completely eliminate variance, thus preserving the necessary dynamic information and allowing its application to time-dependent tasks.

Table 8: Ablation study results of the proposed method on neuromorphic speech recognition dataset SHD.

| Method | Accuracy (%) |
|---|---|
| Baseline (DH-LIF (Zheng et al., 2024)) | 89.86 |
| +Smooth | $90.33_{+0.47}$ |
| +Guidance | $90.46_{+0.60}$ |
| +Both | $\mathbf{91.19}_{+1.33}$ |

Ablation experiments on the 3D point cloud classification dataset ModelNet10 using the PointNet++ architecture and with a timestep of 2 are shown in Table 9, and the results further confirm the effectiveness of our method. We also compare the performance when trained with T=2 but with inference T=1. Our method is able to achieve 94.39% accuracy in 1 timestep inference, surpassing the performance of the vanilla SNN by 1.78% for the same training and inference timestep.

Table 9: Ablation study results of the proposed method on 3D point cloud classification dataset ModelNet10.

| Method | Inference T=2 Acc. (%) | Method | Inference T=1 Acc. (%) |
|---|---|---|---|
| Baseline | 93.75 | Direct training | 91.62 |
| +Smooth | $94.23_{+0.48}$ | Vanilla SNN Training T=2 | 92.61 |
| +Guidance | $94.05_{+0.30}$ | - | - |
| +Both | $\mathbf{94.54}_{+0.79}$ | Ours SNN Training T=2 | **94.39** |

## A.7 OBJECT DETECTION EXPERIMENT

We perform object detection on PASCAL VOC 2012 and COCO 2017 to evaluate whether the proposed method can provide performance gains on non-classification tasks. When our method is used for non-classification tasks (e.g., regression), we can compute the MSE loss between outputs instead of the KL divergence. We take SpikeYOLO (Luo et al., 2024) as the baseline and compute the guidance loss using MSE for its predicted coordinates, and the results for the PASCAL VOC 2012 and COCO 2017 datasets are shown in Table 10 and Table 11, respectively (The training setup follows (Luo et al., 2024) and a total of 80 epochs are trained). The results show that our method can be applied to the object detection task and improve the performance of the baseline model, and should be similarly applicable to other regression tasks.

Note that SpikeYOLO (Luo et al., 2024) uses multi-bit neurons during training so that it can achieve satisfactory performance at lower timesteps. However, this slightly affects the temporal correlation of the SNN. Therefore, our method may be able to achieve more significant facilitation when used in other SNNs (using vanilla single-bit multi-timestep neurons).

Table 10: Comparative results with existing methods on PASCAL VOC 2012. $T \times D$ indicates that $T$ timesteps are set and each timestep is expanded $D$ times. $D$ is set to 1 by default in the comparative methods.

| Method | #Param (M) | $T \times D$ | mAP@50 (%) | mAP@50:95 (%) |
|---|---|---|---|---|
| SpikeYOLO (Luo et al., 2024) | 13.2 | $2 \times 4$ | 48.0 | 30.9 |
| **Ours** | 13.2 | $2 \times 4$ | **48.7** | **31.5** |

Table 11: Comparative results with existing methods on COCO 2017 val. $T \times D$ indicates that $T$ timesteps are set and each timestep is expanded $D$ times. $D$ is set to 1 by default in the comparative methods.

| Method | #Param (M) | $T \times D$ | mAP@50 (%) | mAP@50:95 (%) |
|---|---|---|---|---|
| Spiking-YOLO (Kim et al., 2020) | 10.2 | 3500 | - | 25.7 |
| EMS-YOLO (Su et al., 2023) | 26.9 | 4 | 50.1 | 30.1 |
| Meta-Spikeformer (YOLO) (Yao et al., 2024) | 16.8 | 4 | 50.3 | - |
| SpikeYOLO (Luo et al., 2024) | 13.2 | $2 \times 4$ | 53.6 | 37.9 |
| **Ours** | 13.2 | $2 \times 4$ | **54.0** | **38.4** |

### A.8 TIMESTEP AND POWER CONSUMPTION ANALYSIS OF SPIKING TRANSFORMER MODEL

The timestep of the SNN is proportional to the training and inference overhead. To balance performance and overhead, the timestep in this paper on the neuromorphic dataset was set to 5. However, since the typical spiking Transformer (Shi et al., 2024) model uses larger timesteps such as 10 and 16 on the neuromorphic datasets, we explore here the performance of our method for the Transformer architecture at different timesteps. To evaluate the influence of the time-step hyperparameter on the performance, we conducted experiments on DVS-Gesture based on the SpikingResformer (Shi et al., 2024) architecture, and the results are shown in Table 12. The results show that even at larger timesteps, our method is still able to improve the performance of the baseline SpikingResformer model, although the effect gradually decreases as the performance of the model saturates.

Table 12: Comparison of performance and power consumption at different timesteps. Experiments were performed on DVS-Gesture with the SpikingResformer architecture.

|  |  | T=4 | T=5 | T=8 | T=10 | T=16 |
|---|---|---|---|---|---|---|
| Acc. (%) | Baseline | 89.93 | 90.63 | 92.89 | 93.58 | 96.99 |
|  | +Smoothing | $91.67_{+1.74}$ | $94.44_{+3.81}$ | $94.32_{+1.43}$ | $94.33_{+0.75}$ | $97.23_{+0.24}$ |
| Power (mJ) | Baseline | 0.3699 | 0.4796 | 0.8119 | 1.0374 | 1.5788 |
|  | +Smoothing | 0.3869 | 0.4874 | 0.8079 | 1.0199 | 1.5769 |

In addition, Table 12 compares the power consumption of our method with that of the baseline model. The calculation of the power consumption follows (Shi et al., 2024) (Power consumption is positively correlated with the number of spikes). The results show that our method only slightly increases the power consumption when the timestep is small, and our method with lower power consumption when the timestep is large. In particular, the low number of spikes of the SNN at low timesteps tends to lead to an inadequate feature representation. Our method generates slightly more spikes at low timesteps to enhance the representation quality and thus the performance. In contrast, when the timestep is large, the vanilla SNN suffers from redundant spikes, and our method reduces the redundant spikes, thus improving performance and reducing power consumption. Overall, the difference in power consumption between our method and the baseline model is negligible, and therefore hardly affects the power consumption of the SNN model.

## A.9   EXPERIMENTS ON STATIC IMAGE DATASETS

In addition to neuromorphic datasets, we also conducted experiments on the static object recognition task to demonstrate the generalizability of our method. The comparative results for static datasets are shown in Table 13, where we achieved 96.16% and 79.22% accuracy for CIFAR10 and CIFAR100, respectively, outperforming the other methods.

Table 13: Comparative results (%) on static datasets. * denotes self-implementation results.

| Method | Architecture | Param (M) | T | CIFAR10 | CIFAR100 |
|---|---|---|---|---|---|
| CLIF (Huang et al., 2024) | ResNet-18 | 11.21 | 4 | 94.89 | 77.00 |
| RMP-Loss (Guo et al., 2023a) | ResNet-19 | 12.54 | 4 | 95.51 | 78.28 |
| NDOT (Jiang et al., 2024a) | VGG-11 | 9.23 | 4 | 94.86 | 76.12 |
| TAB (Jiang et al., 2024b) | ResNet-19 | 12.54 | 4 | 94.76 | 76.81 |
| SLT-TET (Anumasa et al., 2024) | ResNet-19 | 12.54 | 4 | 95.18 | 75.01 |
| Spikformer (Zhou et al., 2023) | Spiking Transformer-4-384 | 9.28 | 4 | 95.19 | 77.86 |
| SpikingResformer (Shi et al., 2024) | SpikingResformer-Ti | 10.79 | 4 | 95.93* | 78.23* |
| SDT (Yao et al., 2023) | Spiking Transformer-2-512 | 10.21 | 4 | 95.60 | 78.40 |
| **Ours** | SpikingResformer-Ti | 10.79 | 4 | **96.16** | **79.22** |

To validate the scalability of our method, we performed experiments on Tiny-ImageNet, ImageNet-Hard (Taesiri et al., 2023), and ImageNet.

The comparative results with existing methods on Tiny-ImageNet are shown in Table 14. Our method achieves an accuracy of 58.04% in only 4 timesteps, surpassing other comparative methods.

Table 14: Comparative results on the Tiny-ImageNet dataset.

| Method | Architecture | T | Accuracy (%) |
|---|---|---|---|
| Online LTL (Yang et al., 2022) | VGG-16 | 16 | 56.87 |
| ASGL (Wang et al., 2023) | VGG-13 | 8 | 56.81 |
| Joint A-SNN (Guo et al., 2023b) | VGG-16 | 4 | 55.39 |
| **Ours** | VGG-13 | 4 | **58.04** |

ImageNet-Hard is slightly smaller in scale than ImageNet, but more challenging. We take (Fang et al., 2023b) as the baseline and employ the same training strategy as (Fang et al., 2023b). The comparative results on ImageNet-Hard are shown in Table 15. The results show that our method can still be effective for this challenging dataset.

Table 15: Comparative results with (Fang et al., 2023b) on the challenging ImageNet-Hard dataset.

| Method | Architecture | T | Top-1 Acc. (%) | Top-5 Acc. (%) |
|---|---|---|---|---|
| PSN (Fang et al., 2023b) | SEW-ResNet18 | 4 | 44.32 | 52.57 |
| **Ours** | SEW-ResNet18 | 4 | **45.89** | **53.16** |

Since ImageNet requires more training resources and time, we show the performance of training 250 epochs in Table 16. The results show that our method is already able to achieve competitive performance even with only 250 epochs of training.

Table 16: Comparative results with existing methods on ImageNet dataset.

| Method | Architecture | T | Accuracy (%) |
|---|---|---|---|
| RecDis-SNN (Guo et al., 2022) | ResNet-34 | 6 | 67.33 |
| RMP-Loss (Guo et al., 2023a) | ResNet-34 | 4 | 65.17 |
| SSCL (Zhang et al., 2024b) | ResNet-34 | 4 | 66.78 |
| TAB (Jiang et al., 2024b) | ResNet-34 | 4 | 67.78 |
| SEW-ResNet (Fang et al., 2021a) | SEW-ResNet34 | 4 | 63.18 |
| **Ours** | SEW-ResNet34 | 4 | **69.03** |

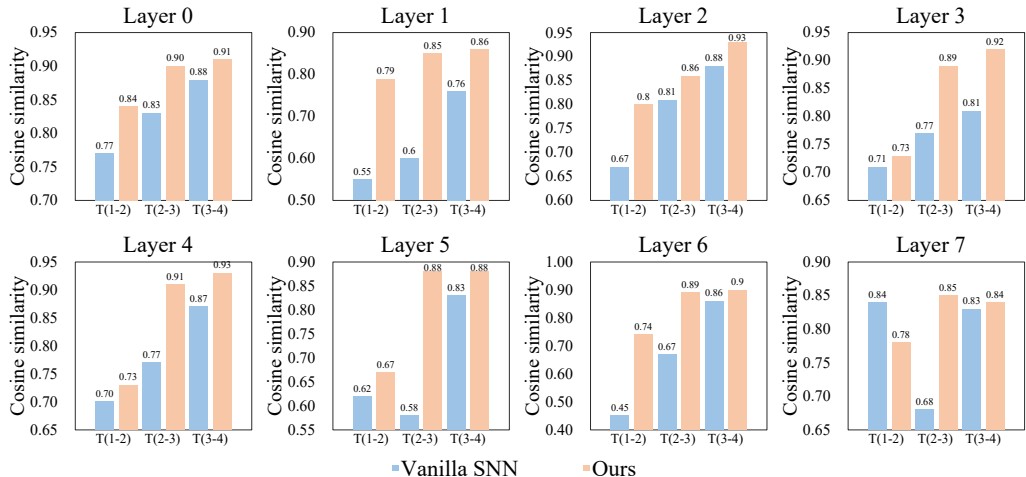

Figure 7: Comparison of the cosine similarity of membrane potential distributions for adjacent timesteps between the vanilla SNN and our method. $T(a - b)$ denotes the similarity of the distribution between timestep $a$ and timestep $b$. Our method significantly improves the similarity of the membrane potential distribution across timesteps, thus facilitating the ensemble performance.

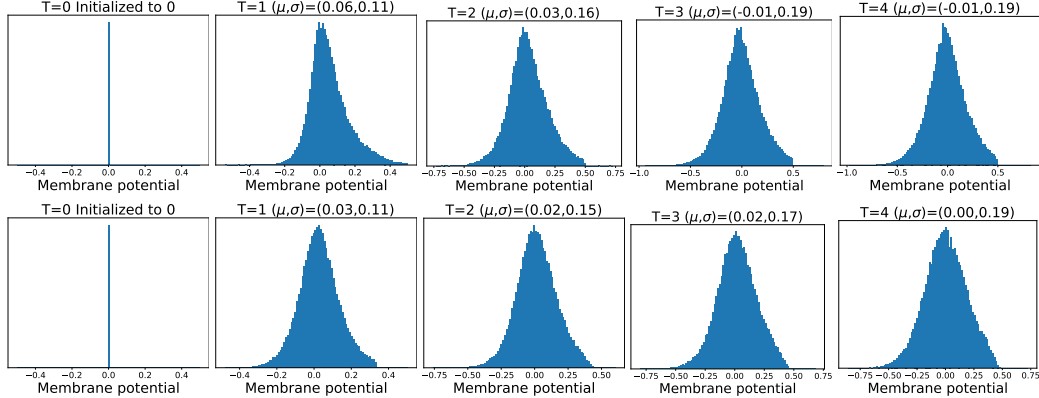

Figure 8: Additional visualization of the membrane potential distribution on CIFAR10-DVS for comparison ($(\mu, \sigma)$ denotes the mean and standard deviation of the distribution). **Top**: Vanilla SNN. **Bottom**: Our method allows for a more stable distribution with smaller differences across timesteps.

## A.10    ADDITIONAL VISUALIZATIONS

In this section, we provide additional visualizations to demonstrate the effectiveness of our method in enhancing the similarity of membrane potential distributions across timesteps.

The cosine similarity of the membrane potential distribution across timesteps for the eight convolutional layer neurons in VGG-9 is shown in Fig. 7. Since the membrane potential is initialized to 0, the similarity of the membrane potential distribution between the 0th timestep and the 1st timestep is 0, and we ignore this term in the figure. It can be seen that our method consistently shows a higher similarity compared to the vanilla SNN, which mitigates the difference in the distribution of membrane potentials across timesteps. Although our method has lower similarity than the vanilla SNN for layer 7 timestep 1 and timestep 2, this exception does not affect our overall role in reducing distributional differences.

In addition, additional visualizations of the membrane potential distribution on CIFAR10-DVS are shown in Fig. 8 to illustrate the effect of our method in smoothing the membrane potential distri-

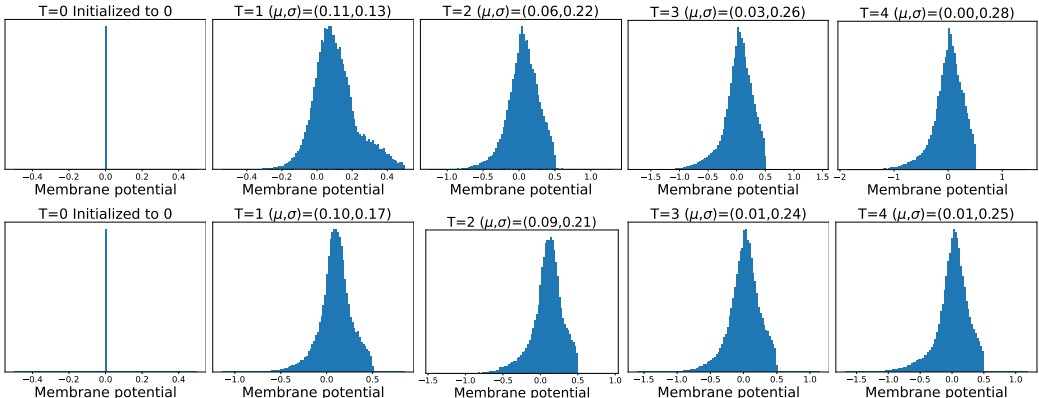

Figure 9: Membrane potential distribution in spiking VGG-9 on DVS-Gesture ($(\mu, \sigma)$ denotes the mean and standard deviation of the distribution). **Top**: Vanilla SNN. **Bottom**: The overall membrane potential distribution of our method is more stable.

bution. The results of the membrane potential visualization on DVS-Gesture are shown in Fig. 9, where again our method shows a more consistent distribution.

