# OpenReview forum: "Rethinking Spiking Neural Networks from an Ensemble Learning Perspective"
_ICLR.cc/2025/Conference — ICLR 2025 Poster_

### Official Review · Reviewer_yRDN · 2024-10-30

**Soundness:** 2
**Presentation:** 3
**Contribution:** 2
**Rating:** 5
**Confidence:** 5

**Summary:**

This paper conducts a thorough investigation into the common issues faced by Spiking Neural Networks (SNNs) and identifies that SNNs primarily suffer from performance degradation due to excessive differences in membrane potential between time steps and uneven feature distribution across time steps. To address these issues, the paper proposes two approaches: **subnetwork guidance** (where each step of the model may be considered as a subnet) and **membrane potential smoothing**. Additionally, ablation experiments are presented to demonstrate the effectiveness of both methods, showing significant improvements in model performance.

**Strengths:**

* The paper is well-written, with a clear and easily understandable motivation. This clarity allows readers to quickly grasp the purpose and significance of the proposed methods.
* The effectiveness of the proposed methods is thoroughly validated on both static image datasets (CIFAR10/CIFAR100) and dynamic vision sensor (DVS) datasets, demonstrating strong performance across different data types.
* Experimental results indicate that the proposed methods are compatible with both CNN-based and Transformer architectures, highlighting their robust generalizability. This versatility suggests that the methods could serve as a widely applicable paradigm in SNN research.

**Weaknesses:**

* The paper mentions the efficiency of Spiking Neural Networks (SNNs) but does not provide experimental evidence on whether the proposed method retains this efficiency. It is recommended to evaluate energy consumption, preferably on hardware platforms such as GPUs, FPGAs, or neuromorphic chips, or to provide theoretical calculations as an alternative. Relevant computation methods can be referenced from prior works [1–4].

* The current experiments are limited to smaller datasets and do not include evaluations on larger, static datasets such as ImageNet. This limitation raises concerns about the method's generalizability and scalability. To strengthen the paper, I suggest including experiments on larger datasets or, at minimum, providing a theoretical analysis of expected performance on large-scale data. Such an evaluation would support the paper's claims of generalizability and effectiveness.

* In Figure 1, the paper shows the membrane potential distribution in a spiking VGG-9 model on the CIFAR10-DVS dataset, comparing the vanilla SNN with the proposed method. Although the figure aims to demonstrate a more stable membrane potential distribution across timesteps with the proposed approach, the visual difference between the two distributions is unclear, making it challenging to assess stability improvements. To clarify this, I recommend including quantitative measures of distribution stability across timesteps, such as mean and variance, or other relevant statistical characteristics. Additionally, a side-by-side comparison of these statistics could highlight any notable differences, making the figure’s intended message more accessible to readers.


[1] Horowitz, M. "1.1 Computing's energy problem (and what we can do about it)." *IEEE International Solid-State Circuits Conference Digest of Technical Papers (ISSCC)*, 2014.

[2] Luo, X. et al. "Integer-Valued Training and Spike-Driven Inference Spiking Neural Network for High-performance and Energy-efficient Object Detection." *arXiv preprint arXiv:2407.20708*, 2024.

[3] Yao, M. et al. "Spike-driven Transformer V2: Meta Spiking Neural Network Architecture Inspiring the Design of Next-generation Neuromorphic Chips." *Twelfth International Conference on Learning Representations (ICLR)*.

[4] Shen, G. et al. "Are Conventional SNNs Really Efficient? A Perspective from Network Quantization." *Proceedings of the IEEE/CVF Conference on Computer Vision and Pattern Recognition*, 2024.

**Questions:**

* The study lacks experiments on large-scale static datasets, such as ImageNet. Can the authors provide additional validation on these datasets to better assess the generalizability of their methods?

* For the spiking transformer model, the commonly used timesteps are 4, 10, or 16. Could the authors clarify the rationale behind selecting a timestep of 5, and discuss any observed impact on performance or efficiency?

* While the paper highlights the efficiency of Spiking Neural Networks (SNNs), there is no evaluation of the proposed method’s energy consumption. Could the authors provide insights into whether this method might lead to high energy consumption, potentially offsetting the efficiency benefits typically associated with SNNs? A practical energy consumption assessment on hardware platforms such as GPUs, FPGAs, or neuromorphic chips would strengthen this aspect.

* In Figure 1, the paper compares the membrane potential distribution of a spiking VGG-9 model on the CIFAR10-DVS dataset, contrasting the vanilla SNN with the proposed method. The figure suggests that the proposed method stabilizes the membrane potential distribution across timesteps. However, the distinction between the two distributions is not clearly evident, making it difficult to interpret the extent of improvement. Could the authors provide further clarification or additional visualizations to better illustrate this difference?

---

### Official Review · Reviewer_eENu · 2024-11-03

**Soundness:** 3
**Presentation:** 4
**Contribution:** 3
**Rating:** 6
**Confidence:** 4

**Summary:**

This work rethinks spiking neural networks (SNNs) from the perspective of an ensemble of multiple temporal subnetworks and argues that improving the stability of the membrane potential and the output of each subnetwork would benefit the performance of SNNs. Based on these motives, this work proposes an approach of membrane potential smoothing and a well-designed learning objective with temporally adjacent subnetwork guidance to improve stability. When applied to several SNNs, the proposed method achieves higher performance across multiple datasets and tasks.

**Strengths:**

1. Most commendable is the match between the motivation and the proposed methodology of this work. The stability of temporal subnetworks is a meaningful research focus, and both membrane potential smoothing and temporally adjacent subnetwork guidance intuitively fit well with the goal of improving stability, especially since the smoothing design is concise and sufficiently novel.
2. The method efficiently improves several SNNs' performance across multiple datasets and tasks, demonstrating its effectiveness and generalizability.
3. The ablation studies are very comprehensive, illustrating the contribution of the methods and the influence of the hyperparameters. The visualization results show graphically the advantages of the method for improving stability.
4. The paper is well-written and easy to follow.

**Weaknesses:**

Major points:
1. The view of SNNs as an ensemble of temporal subnetworks actually originates from Spiking PointNet [1], and they proposed this view because the point cloud data is static and does not involve time dependence. One of the great features of SNNs is their ability to handle temporal data. Therefore, it is doubted whether the ensemble view is still justified when SNNs process data with strong time dependence, and whether membrane potential smoothing is necessary, as more varied potentials seem to be more conducive to representing the dynamic information of the input.
2. The idea of promoting the consistency of SNN instances at different timesteps has been proposed in previous work [2]. While they used contrastive learning to bring network features closer at different time steps, this work uses KL divergence to bring the probability distribution of outputs closer at different time steps. The former design has broader applicability, while the latter is limited to classification tasks.
3. Given that all experiments in this work are performed on classification tasks, the lack of ImageNet results makes the method somewhat unconvincing.

Minor point:
1. The statement that membrane potential smoothing is effective in other neurons is reckless (lines 278-279) and there is a lack of experiments to prove this.

[1] Dayong Ren, Zhe Ma, Yuanpei Chen, Weihang Peng, Xiaode Liu, Yuhan Zhang, and Yufei Guo. Spiking pointnet: Spiking neural networks for point clouds. In Conference on Neural Information Processing Systems. 2023.

[2] Haonan Qiu, Zeyin Song, Yanqi Chen, Munan Ning, Wei Fang, Tao Sun, Zhengyu Ma, Li Yuan & Yonghong Tian. Temporal Contrastive Learning for Spiking Neural Networks. In Conference on Artificial Neural Networks and Machine Learning. 2024.

**Questions:**

1. How does the method perform when applied to some more time-dependent datasets, such as spiking Heidelberg digits (SHD) [3]? Does membrane potential smoothing still benefit the performance in such cases?
2. How does temporally adjacent subnetwork guidance compare to or improve upon the contrastive learning method [2]? In addition, how can the loss function be adapted for non-classification tasks, such as object detection which requires the output of coordinates?
3. Please provide results on ImageNet or explain why ImageNet experiments are not included.

[3] Benjamin Cramer, Yannik Stradmann, Johannes Schemmel, and Friedemann Zenke. The Heidelberg Spiking Data Sets for the Systematic Evaluation of Spiking Neural Networks. IEEE Transactions on Neural Networks and Learning Systems. 2020.

---

> ### Comment · Reviewer_eENu · 2024-11-26
>
> Thanks for doing these additional experiments, which address my concerns to a great extent. However, most of the performance gains are marginal and derived from the baseline performance. Therefore, I decide to maintain my score.

---

### Official Review · Reviewer_Szjy · 2024-11-03

**Soundness:** 3
**Presentation:** 3
**Contribution:** 2
**Rating:** 6
**Confidence:** 3

**Summary:**

The authors propose to adaptively smooth the membrane potential cross adjacent timesteps to reduce the initial state discrepancy.  They propose the temporally adjacent subnetwork guidance to improve the output consistency of subnetworks through distillation. Compared to existing methods, their method shows superior performance in neuromorphic/static object/gesture recognition and 3D
point cloud classification tasks.

**Strengths:**

***1.*** This paper is well-structured with high-quality figures, tables, and formulas.

***2.*** The paper includes relatively adequate analysis, comparative experiments, and ablation studies.

**Weaknesses:**

***1.*** This paper is not the first to view SNNs as a form of ensemble learning; it should reference prior works that have held this perspective.

***2.*** One of the motivations of this paper is to make instances at each time step more consistent; however, this has been demonstrated in many previous works and cannot be considered a novel contribution. But the method is novel. Author should change the focus when the write the paper.

**Questions:**

***1.*** The text in the figures is too small; please enlarge it to improve readability.

***2.*** Why perform 3D point cloud classification experiments. How does this validate the method's characteristics and effectiveness?

***3.*** The authors' method aims for the membrane potential distribution to become more consistent over time. However, at timestep 0, the method still shows a significant difference compared to other timesteps, with this difference being greater than that between other timesteps. This discrepancy makes it difficult for the method to support the argument of achieving consistent membrane potentials.

---

> ### Comment · Reviewer_Szjy · 2024-11-26
> **Thanks for authors**
>
> Thanks for authors' rebuttal. It is more clear now.

---

### Official Review · Reviewer_rfBg · 2024-11-03

**Soundness:** 2
**Presentation:** 2
**Contribution:** 2
**Rating:** 5
**Confidence:** 4

**Summary:**

The authors proposed a method to improve the performance of SNN from a new perspective, ensemble learning, by membrane potential smoothing and temporally adjacent subnetwork guidance. The smoothing method solved the problem that if the differences between initial states of neuron membrane potentials is excessive, the performance of SNN in ensemble learning will be degraded. The knowledge distillation in temporally adjacent subnetworks also improves the consistency of the subnetworks’ output.

**Strengths:**

1.The proposed membrane potential smoothing not only mitigates the difference in membrane potential distribution, which degrades the performance in ensemble learning of SNN, but also reduces the influence of the temporal gradient vanishing.

**Weaknesses:**

1.Some spelling errors. In ABSTRACT, line 25, “Transofrmer” -> “Transformer”.

2.The differences in the membrane potential distribution may not be easily observed. Adding quantitative data tables may be better.

3.The proposed method does not change the core of previous methods, the novelty compared with existing architectures and neuron models should be explained further, and how to improve the overall performance by these proposed approaches.

4. To be honest, the performance of the proposed model on CIFAR10-DVS and DVS-GESTURE is not higher compared to most of the existing spiking models (proposed in the last six months) based on VGG or ResNet backbone, Thus I could not determine the novelty and effectiveness of the proposed model.

**Questions:**

1.The proposed method does not change the core of previous methods, the novelty compared with existing architectures and neuron models should be explained further, and how to improve the overall performance by these proposed approach.

---

> ### Author Response · Authors · 2024-11-21
> **Responses to comments (Part 2)**
>
> * **Generalizability of the proposed method.** We smooth neurons without changing their core dynamics, allowing our method to be applied seamlessly to existing diverse neurons such as LIF, PLIF [1], and CLIF [2]. Furthermore, our training strategy does not change the network architecture and can therefore be applied to different SNN architectures, such as the VGG, ResNet, and Transformer architectures on which we demonstrated its effectiveness in the experiments of the paper. Thus, by not changing the core of existing neurons or architectures, our method is generalizable and can be easily applied to more advanced neuron models and architectures in the future. We believe that such innovations can contribute to the long-term research of the SNN community. The following table demonstrates the effectiveness of our method on LIF, CLIF [2], and PLIF [3] neurons on CIFAR10-DVS.
> | Method | LIF | CLIF [2] | PLIF [3]|
> |-------- |-----|-----|-----|
> | Baseline| 73.97| 74.97| 74.83|
> | +Smoothing | 74.80(+0.83)  | 75.97(+1.00)| 75.10(+0.27)|
>
>     * We also combined membrane potential smoothing with DH-LIF [4] on the temporal SHD dataset, and the results in the following table again show performance gains. We hope this demonstrates the compatibility of our method with existing methods, as well as its superior generalization.
>     | Method |SRNN-LIF | DH-LIF [4] |
>     |-------- |-----|-----|
>     | Baseline| 81.87| 89.86|
>     | +Smoothing | 83.04(+1.17) | 90.33(+0.47)|
>
> * **The interpretability of the proposed method.** Our method is primarily designed to promote the stability of the SNN ensemble across timesteps, which is demonstrated by visualizing the membrane potential distribution and the timestep-wise output. By adding quantitative data to the visualized membrane potential, it can be seen that our method significantly improves the stability of the membrane potential. In addition, the timestep-wise outputs are visualized in Fig. 4 of the paper, and it can be seen that our method makes the outputs of the SNN more stable at each timestep, while the outputs of the vanilla SNN are highly variable (especially at timestep 0 and other timesteps).
>
> * **Comparison with recently proposed spiking neurons.** The following table shows the results of our method on CIFAR10-DVS and DVS-Gesture compared to neurons published in the last six months. It is worth noting that we train the VGG-9 architecture for only 100 epochs, while other methods are more cumbersome to train (e.g., [6] trains for 320 epochs and uses data augmentation). Therefore, we additionally trained our model with strategy 2 (200 epochs of training with data augmentation), and the results show that the performance is further improved.
>
> | Method |Architecture | T | CIFAR10-DVS | DVS-Gesture|
> |-------- |-----|-----|-----| -----|
> | CLIF (ICML'24) [2]| VGG-9| 5 | 74.97 | 91.55|
> | Ternary Spike (AAAI’24) [5]| ResNet-20 | 10 | 78.70 | -|
> | Progressive Training+LM-H model (ICLR’24) [6]| ResNet-19 | 10 | 79.10 | -|
> | Ours | VGG-9  | 5 | 76.77| 93.23 |
> | Ours (strategy 2) | VGGSNN | 4 | 83.20 | -|
>
>    In addition to neuromorphic object recognition, we also compared other latest neurons in the temporal SHD dataset, and the results are tabulated below:
>
> | Method |Architecture | Params | SHD Acc. (\%)|
> |-------- |-----|-----|-----|
> | DH-LIF (Nature Communications 2024) [4] | 2-layer,8branch | 0.05M | 89.86|
> | TC-LIF (AAAI’24) [7] |recurrent | 0.14 | 88.91|
> | Ours |  2-layer,8branch | 0.05M | 90.33|
>
>   We hope that these comparative results will demonstrate the effectiveness and performance advantages of our method.
>
>
> ```
> [2] CLIF: Complementary Leaky Integrate-and-Fire Neuron for Spiking Neural Networks. In ICML. 2024.
> [3] Incorporating Learnable Membrane Time Constant to Enhance Learning of Spiking Neural Networks. In ICCV. 2021.
> [4] Temporal Dendritic Heterogeneity Incorporated with Spiking Neural Networks for Learning Multi-Timescale Dynamics. Nature Communications. 2024.
> [5] Ternary Spike: Learning Ternary Spikes for Spiking Neural Networks. In AAAI. 2024.
> [6] A Progressive Training Framework for Spiking Neural Networks with Learnable Multi-hierarchical Model. In ICLR. 2024.
> [7] TC-LIF: A Two-Compartment Spiking Neuron Model for Long-term Sequential Modelling. In AAAI. 2024.
> ```

---

### Meta-Review · Area_Chair_Rfjs · 2024-12-19

**Metareview:**

This paper reinforces the ensemble of multiple temporal subnetworks nature of Spiking neural networks and identifies challenges in using such ensemble - instability due to initial states. To mitigate this instability, the paper proposes membrane potential smoothing across multiple timesteps that also mitigates temporal vanishing gradient.

The stability of temporal subnetworks is a meaningful research focus, and both membrane potential smoothing and temporally adjacent subnetwork guidance intuitively fit well with the goal of improving stability, especially since the smoothing design is concise and sufficiently novel.[Reviewer eENu]

Weakness:
While the results show some improvement, the gains compared to baselines aren’t very significant.[Reviewer yRDN, Reviewer rfBg]
Novelty is incremental as indicated by other reviewers.
Impact of the proposed work is not very evident - how does this impact the applicability of SNN in real-world applications and/or how does this impact the research community?

**Additional Comments On Reviewer Discussion:**

Authors have addressed most of reviewer's concerns. Yet, as indicated by reviewers, the performance gains are not significant in comparison to baselines as agreed by Reviewers yRDN and rfBg (although Reviewer rfBg increased the score).

Authors argue that the Ternary Spike [1], one of the baselines, use double the latency and a heavier network architecture (ResNet-20 compared to the author's VGG-9) and these factors make the performance improvement of the proposed method seem limited. They also claim that given the same architecture and latency, the proposed method offers significantly better performance than existing methods - which is not validated [see response to reviewer rfBg].

---

### Decision · Program_Chairs · 2025-01-22

Accept (Poster)